# Research on the Reliability of Advanced Packaging under Multi-Field Coupling: A Review

**DOI:** 10.3390/mi15040422

**Published:** 2024-03-22

**Authors:** Yongkun Wang, Haozheng Liu, Linghua Huo, Haobin Li, Wenchao Tian, Haoyue Ji, Si Chen

**Affiliations:** 1State Key Laboratory of Electromechanical Integrated Manufacturing of High-Performance Electronic Equipments, Xidian University, Xi’an 710071, China; 22041212954@stu.xidian.edu.cn (H.L.); 21041211815@stu.xidian.edu.cn (L.H.); 21041211857@stu.xidian.edu.cn (H.L.); wctian@xidian.edu.cn (W.T.); 2The 58th Research Institute of China Electronics Technology Group Corporation, Wuxi 214000, China; cameljihy@126.com; 3The Fifth Electronics Research Institute of Ministry of Industry and Information Technology, Guangzhou 510000, China; chensiceprei@yeah.net

**Keywords:** advanced packaging, multi-field coupling, micro-bump, TSV

## Abstract

With the advancement of Moore’s Law reaching its limits, advanced packaging technologies represented by Flip Chip (FC), Wafer-Level Packaging (WLP), System in Package (SiP), and 3D packaging have received significant attention. While advanced packaging has made breakthroughs in achieving high performance, miniaturization, and low cost, the smaller thermal space and higher power density have created complex physical fields such as electricity, heat, and stress. The packaging interconnects responsible for electrical transmission are prone to serious reliability issues, leading to the device’s failure. Therefore, conducting multi-field coupling research on the reliability of advanced packaging interconnects is necessary. The development of packaging and the characteristics of advanced packaging are reviewed. The reliability issues of advanced packaging under thermal, electrical, and electromagnetic fields are discussed, as well as the methods and current research of multi-field coupling in advanced packaging. Finally, the prospect of the multi-field coupling reliability of advanced packaging is summarized to provide references for the reliability research of advanced packaging.

## 1. Introduction

Microelectronic packaging technology plays a vital role in semiconductor manufacturing. Its primary function is to provide protection and ensure the product’s safety. This involves establishing electrical connections between the bare chip and external circuits using wire bonding, micro-bumps, and others. The assembly is then encapsulated with a casing designed to safeguard the internal circuitry.

Figure 1 depicts the evolution of electronic packaging [1]. In the 1970s, chip packaging predominantly consisted of through-hole devices and plug-ins, primarily employed in manual soldering assemblies, such as the Dual In-line Package (DIP). In the 1980s, surface-mount soldering technology emerged, featuring flat pins on two sides or all around the component, as exemplified by the Small Outline Package (SOP) and Quad Flat Package (QFP). By the 1990s, the increasing complexity of single-chip functionality and higher input/output density posed challenges for distributing pins solely around the package’s perimeter. This gave rise to packaging forms exemplified by Ball Grid Array (BGA) and Flip Chip (FC) technologies, which utilized the entire backside area of the chip. The inception of the Chip-Scale Package (CSP) in the early 21st century further enhanced chip area utilization. Post-2010, the advancement of emerging device technologies, packaging integration, and miniaturization reached the System in Package (SiP) level. An SiP integrates passive components and active devices and incorporates various functional devices within a single package, such as MEMS, biochips, sensors, etc. The development of 3D packaging, such as chip stacking, further propelled the progress of SiP.

As time progresses, the number of I/O pins and the integration density in packaging continues to rise. In recent years, the rapid development of electronic information and the emergence of automotive electronics, 5G technology, and artificial intelligence have fueled the demand for high-performance chips. High computational power necessitates large-scale chip integration, while high-memory-bandwidth processing requires high packaging I/O density and low-electrical-loss interconnects between devices. The advent of 5G and foldable smartphones also calls for thinner and smaller form factors [2]. Due to low efficiency and performance, traditional packaging technologies that rely on wire bonding as the dominant interconnect technique needed to be improved to meet these demands. Consequently, advanced packaging methods utilizing point-to-point interconnects emerged.

Advanced packaging primarily refers to technologies such as FC, Wafer-Level Packaging (WLP), SiP, and 3D packaging. Compared to traditional packaging, advanced packaging offers higher packaging density and integration, allowing for the incorporation of more functionalities within a single package. SiP and Package on Package (PoP) technologies have laid the foundations of advanced packaging. The introduction of 2D integration technologies like WLP, FC, and 3D packaging, along with through-silicon via (TSV) technology, further reduces the interconnect distances between chips. These advancements will drive the progress of advanced packaging in the future.

Interconnect technology plays a critical role in advanced packaging. It not only influences the performance and reliability of the packaging but also directly affects essential metrics such as system performance and power consumption. The primary forms of interconnects in advanced packaging include micro-bumps, through-silicon via, and Redistribution Layer (RDL) technologies, as depicted in Figure 2.

Micro-bumps are primarily employed for chip-to-substrate connections due to high reliability and low-cost advantages. However, micro-bumps work in complex and variable environments, including high temperatures, thermal cycles, mechanical vibrations, and currents. Among the factors influencing micro-bumps’ reliability, electromigration and thermal migration are the main concerns [3]. TSV technology is a vertical interconnect technique that involves creating vertical openings in the chip and filling them with conductive material. This enables the transmission of thermal and electrical signals within the chip, significantly enhancing chip integration and performance. TSV encounters challenges such as differences in the coefficient of thermal expansion between filling materials and adjacent materials, resulting in thermal mismatch issues and interface cracking or delamination under thermal stress [4,5]. Furthermore, parasitic effects in TSV structures in high-frequency environments can impact signal transmission efficiency [6]. Redistribution Layer (RDL) technology is a horizontal interconnect technique. An RDL is a layered structure that connects chip pads to external package pads, typically positioned between the chip and the package substrate. However, the connection between the metal lines in the RDL layer and the chip is established through small solder balls or pillars, which can be vulnerable to failure due to thermal stress or other factors [7,8].

The evolution of advanced packaging technology has enabled chips to reach higher levels of integration and performance. However, this advancement is accompanied by increased structural and environmental complexity. During device operation, various energies, including mechanical forces, thermal energy, electrical energy, and magnetic energy, are generated, converted, and transferred across different physical fields. These energies interact, collectively influencing the packaging structure. As a result, the package’s internal structures may experience various forms of failure, such as fatigue fractures, stress concentrations due to thermal expansion, and electromagnetic interference. The interconnects are pivotal in establishing electrical connectivity within the package, significantly influencing its overall reliability.

Therefore, to improve packaging reliability and optimize packaging design, it is essential to conduct a multi-physical-field coupling analysis focused on structures of advanced packaging like bumps or TSVs. Multi-physical-field coupling analysis examines the interactions among various physical fields, including mechanical, thermal, electrical, and magnetic fields, and the properties of materials like thermal conductivity, electrical conductivity, and magnetic permeability. This analysis involves developing suitable mathematical and physical models to calculate multiple physical fields’ mutual influences and coupling effects.

Recent research on the reliability of advanced packaging under thermal, electrical, and electromagnetic fields is summarized in Section 2. It lists failures and measures taken for advanced packaging devices in each physical field. The computational methods for different physical field coupling models are reviewed in Section 3. Additionally, recent research on multi-field coupling reliability in advanced packaging is reviewed. Section 4 provides an outlook on developing multi-field coupling reliability in advanced packaging.

## 2. Overview of Electronic Packaging Reliability

To examine the reliability of advanced packaging under multi-field coupling, examining each physical field’s influence on the package’s reliability is necessary. In traditional packaging, thermal stress issues dominate, often leading to device failures. For instance, materials with different coefficients of thermal expansion (CTEs) experience thermal mismatch at high temperatures, leading to fatigue and fracture in the vulnerable parts of the device during temperature cycling, and so on. As for the reliability issues in advanced packaging devices, in addition to thermal problems and related electrical problems, the crosstalk problems caused by electromagnetic coupling in high-density interconnects cannot be ignored. This chapter aims to delineate the mechanisms of various physical field effects in advanced packaging and their associated challenges.

### 2.1. Thermal Reliability

Thermal reliability in advanced packaging can be categorized into two main areas: reliability under high-temperature thermal fields and reliability under thermal cycling. The former relates to the reliability of electronic devices during operation when the temperature rises due to power consumption. Advanced packaging, characterized by significantly increased density compared to traditional packaging, consists of multi-layer structures. The limited area poses challenges in effectively dissipating the generated heat, leading to mechanical deformation like warpage or cracks. Moreover, high temperatures can activate thermal energy, causing the migration of atoms or ions within the materials, which can impact the structure and properties of the materials. On the other hand, reliability under thermal cycling pertains to the thermal stress cycles between materials with different CTEs during the manufacturing or service process of packaged devices, resulting from periodic temperature variations in the environment. This can lead to fatigue or cracking in packaging structures.

The high-temperature thermal field is the primary cause of most reliability problems in packaging. In advanced packaging, chips are commonly mounted on substrates, and interconnects are established through TSVs and bumps. The materials within the package, each with a different CTE, undergo varying expansion levels due to temperature changes. As a result, thermal mismatch occurs at bonding interfaces or solder joint regions, leading to structural failures such as fracture and cracking within the packaging.

Thermal mismatches can cause the warpage of the packaging substrate, which can lead to open circuits between the bumps and the substrate, as shown in Figure 3. Previous studies have demonstrated that warping can be improved by optimizing parameters such as the substrate, underfill material, molding compound, and chip dimensions. Tsai conducted measurements and simulations on a large number of FC BGA packages and found that thinner core substrates often lead to worse warpages. Low-CTE materials for the substrate and the underfill can effectively reduce warpage [9]. Hou investigated the thermal-induced warpage during the molding process of fan-out packages and found that higher CTEs and thicker molding compounds can also reduce warpage [10]. Additionally, the ratio of chip size to package size can affect warpage, with smaller chip sizes resulting in less warpage under the same mold conditions [11].

The thermal stress caused by high-temperature thermal fields can also lead to interconnecting cracking. As shown in Figure 3, the interconnect structure with TSVs involves filling conductors (typically Cu) in the silicon interposer, surrounded by dielectric layers. Influenced by thermal mismatch, gaps and delamination can quickly form between the sidewalls of these structures. Moreover, the increase in the height of Cu pillars at high temperatures can lead to cracking of the dielectric layer, resulting in mechanical failures [12]. Additionally, in interconnect structures containing both TSVs and bumps, the junction between the TSVs and bumps is prone to experiencing the highest stress levels [13]. Therefore, it is essential to avoid manufacturing defects such as poor interface bonding and roughness during fabrication to prevent thermal mismatch.

The reliability issues caused by thermal cycling should be considered. The product undergoes multiple steps during manufacturing such as bonding, reflow, and injection molding. During service, temperature variations between room and operating temperatures are also encountered. For interconnect structures in advanced packaging, such as FC bumps, there is a risk of fatigue failure under such temperature cycling. The solder joints’ low-cycle fatigue failure response is a creep-fatigue mechanism that involves crack initiation and crack growth until complete rupture of the solder connection [14], as shown in Figure 4. Underfill materials at the bottom can effectively enhance the thermal fatigue life of the bumps. The thermal fatigue life of the bumps is closely related to the material properties of the underfill. A higher modulus and lower CTE of the underfill can significantly enhance the thermal fatigue life of the bumps [15]. The shape of the bumps also affects the fatigue life. Increasing the volume and reducing the diameter can improve the fatigue life [16], while uneven protrusions can reduce the fatigue life [17].

Most internal thermal problems in advanced packaging are caused by CTE mismatch among the internal materials, and many researchers have conducted studies to optimize CTE matching. Banijamali proposed a method for interconnect routing in a 28 nm chip package using TSV interposer structures, effectively reducing CTE mismatch [19]. XJ Fan found that the polymer structure in the bumps can significantly improve the thermal–mechanical reliability of WLP. This is because the polymer film in the silicon and bumps acts as a buffer, reducing the stress in the bumps [20]. Li presented a coaxial TSV structure with an embedded cooling cavity in the silicon interposer. This configuration achieves excellent electrical transmission and enhances heat dissipation, effectively reducing thermal stress [21].

### 2.2. Electrical Reliability

Electric fields serve as the source of Joule heating. In addition to causing the thermal problems mentioned earlier, electric fields also give rise to electromigration in conductive materials. Electromigration refers to the phenomenon where atoms in a conductor material move along the direction of the electric field induced by the interaction between electrons and atoms. It is a diffusion behavior driven by a driving force [22].

Electromigration is a common reliability problem in interconnects. Metal atoms in the conductor undergo migration under high current density, accumulating stress in regions where the net atomic flux is depleted. As the local stress reaches a critical tensile stress level, it can eventually result in void formation and open interconnect circuits. The formation of voids also leads to current crowding, causing an increase in resistance. In regions with a net accumulation of atoms, the local stress becomes increasingly compressive, which can result in metal extrusion. A short circuit occurs when the extruded metal comes into contact with adjacent interconnects or vias [23]. With the increasing packaging density and size reduction, the dimensions of metal interconnects reach the nanoscale, and the current density approaches the maximum limit for electromigration reliability. Therefore, improving electromigration reliability in high-density and small-sized interconnects is meaningful. Li used in situ-grown graphene as a capping material, which improved the reliability of 80 nm, 100 nm, and 120 nm wide copper interconnects [24].

For solder bumps, high current density accelerates the reaction of intermetallic compounds (IMCs) between metals. This leads to increased thickness of IMCs at the anode of the solder joint and the formation of voids and cracks at the cathode [25], as shown in Figure 5. Additionally, the Joule heating generated by current crowding creates a temperature gradient on both sides of the solder bump, resulting in thermal migration [26]. The combined effects of electromigration and thermal migration accelerate the failure. Existing research has identified several methods to improve the failure of solder bumps induced by current crowding. One approach is to optimize the interconnect structure and use different-shaped solder bumps [27]. Other methods involve reducing the pad opening size and bump height [28] and optimizing the layout of the solder bump array. From the materials’ perspective, microalloying and particle reinforcement of the solder bump metals, as well as the design of surface coatings on the substrate, can be employed to improve their performance [29].

Research on electromigration in TSVs has been relatively limited, as it has been generally believed that the current density in the vias is much lower than that in the interconnect metals. However, Tan pointed out that electromigration can occur in TSVs, with temperature gradients and mechanical stresses replacing current density as the primary driving forces [31]. Figure 6 illustrates electromigration voids at the top, middle, and bottom of a TSV. Optimizing electromigration in TSVs requires not only controlling current density, such as using larger-diameter TSVs, introducing buffer layers or substrate processes, or employing current balancing circuits or current distribution networks to achieve more uniform current distribution [32,33], but also focusing on improving temperature gradients and stresses. For example, when designing TSVs, it is crucial to avoid stress gradients caused by structural dimension mismatches [34].

### 2.3. Electromagnetic Reliability

With the increase in integration levels and operating frequencies, coupled with the trend of chip stacking in the direction of 2.5D and 3D, the electromagnetic environment surrounding integrated circuits is becoming more intricate. As a result, there is a growing concern for signal integrity issues. Challenges such as electromagnetic interference and coupling within the package pose significant obstacles to the design of advanced packaging. These issues must be carefully addressed to ensure optimal performance and reliability of the integrated circuits in the package.

In the stacked structure of 3D IC integration, TSVs may bring impedance variations due to the signal’s transition through vertical connections across silicon wafers, which involve different materials and geometric structures. Under various stresses, the materials and geometric structures may experience alterations, leading to impedance mismatches. These impedance variations can cause signal reflections and oscillations, impacting the overall signal integrity in the system. To mitigate signal transmission problems caused by impedance variations, it is possible to optimize the geometry and structure of TSVs by properly designing parameters such as radius size, cross-sectional shape, and TSV inclination angle [36].

The skin effect and proximity effect are also significant factors affecting interconnect transmission performance [37]. The skin effect refers to the non-uniform distribution of current density near the surface of a conductor at high frequencies. In contrast, the proximity effect refers to the mutual influence between adjacent interconnects. These effects can lead to non-uniform current distribution on the interconnects, affecting signal transmission performance. Additionally, the surface roughness generated during the fabrication increases the effective conductor area in the surface region, thereby increasing the conductor loss caused by the skin effect [38].

The spacing between signal lines becomes narrower with the increase in the integration density. Due to the temperature sensitivity of parasitic parameters in interconnect structures such as TSVs [39], the thermal–electrical coupling within the package can cause variations in these parasitic parameters, leading to exacerbated crosstalk issues. Crosstalk refers to the interference caused by the electromagnetic field of one signal line on adjacent signal lines. Various isolation and shielding techniques can be employed to mitigate crosstalk. For example, designing coaxial TSV structures with a central signal transmission conductor surrounded by concentric ground loops can effectively reduce signal loss and coupling noise (shown in Figure 7b) [40]. Adjusting the shape and size of protrusions, such as tapered protrusions, can also help to improve crosstalk delay due to their lower capacitive coupling and smaller volume fraction [41,42]. Additionally, incorporating shielding structures, such as adding ground shielding rings around TSVs (shown in Figure 7a) [43], introducing ground interconnect arrays (shown in Figure 7c) [44], or implementing appropriate grounding in the package enclosure [45], can further mitigate crosstalk.

The physical field mentioned above is closely related to packaging reliability. In practical situations, they are not independent of each other but rather mutually influential. Changes occurring in one physical field can trigger changes in other physical fields, significantly increasing the possibility of failure. Conducting multi-physics coupling analysis is crucial for comprehending the underlying problems and enables targeted optimizations.

## 3. Advanced Packaging Multi-Field Coupling Research Progress

### 3.1. Overview of Multi-Field Coupling Methods

As discussed in Section 2, failures in packaging primarily stem from thermal, electrical, and electromagnetic issues. Extensive research has been dedicated to the individual calculation of these physical fields. However, to study the reliability of packaging, it is essential to address the distribution issues of thermal, electrical, and electromagnetic fields and consider the intercoupling among these multi-physical fields.

Determining the coupling relationships between these physical fields affects the calculation’s cost and efficiency and the results’ accuracy. The coupling relationship between electrical, thermal, and mechanical analysis is illustrated in Figure 8, which includes several standard physical fields in packaging.

In practical research, it is essential to devise distinct solutions tailored to specific problems. Advanced packaging encompasses a broad concept, and its reliability issues can be categorized into numerous sub-areas. At their core, these issues primarily revolve around thermal management, mechanical stress, and electrical performance. A detailed explanation of the intercoupling relationships and calculation methodologies between various physical fields is provided. This discussion also encompasses the research objectives of advanced packaging and the methods pertinent to these objectives.

#### 3.1.1. Thermo-Mechanical Coupling

The fundamental concept of thermal–mechanical coupling involves several key steps. Initially, the temperature distribution across the system is established by solving the heat conduction equation, incorporating relevant boundary conditions. Subsequently, using the thermal expansion coefficients and the derived temperature distribution, the thermal expansion of each component in response to temperature fluctuations is calculated. Finally, the thermal stress distribution is determined using the stress–strain relationship in conjunction with the calculated thermal expansions.

The governing equations for thermal–mechanical coupling calculation primarily consist of the strain equation and the heat conduction equation. The strain equation describes the deformation behavior under the action of external forces. The total strain is composed of mechanical strain and thermal strain induced by temperature changes, and its expression is as follows:(1)$$\varepsilon_{\mathrm{total}}=\varepsilon_{m}+\varepsilon_{t}$$
(2)$$\sigma =C\varepsilon_{m}$$
(3)$$\varepsilon_{t}=\alpha \Delta T$$

Equation (1) represents the total strain, *ε_total_*, which is composed of mechanical strain (*ε_m_*) obtained using the generalized Hooke’s law (Equation (2)), and thermal strain (*ε_t_*) induced by temperature changes. The stress tensor (*σ*) is related to the strain through the stiffness matrix (*C*).

Equation (3) describes the thermal expansion (*ε_t_*) caused by temperature variations. It is obtained using the thermal expansion equation, where *α* is the thermal expansion coefficient and Δ*T* represents the temperature difference.

The heat conduction equation describes the heat transfer behavior within an object or structure and is expressed as follows:(4)$$C_{p}\rho \frac{\partial T}{\partial t}=\lambda \left(\frac{\partial^{2}T}{\partial x^{2}}+\frac{\partial^{2}T}{\partial y^{2}}+\frac{\partial^{2}T}{\partial z^{2}}\right)+q_{v}$$
in which *C_p_* represents the specific heat capacity, *ρ* represents the density, *q_v_* represents the heat generation per unit volume, *T* represents the temperature, and *t* represents the time.

In some instances, it is necessary to consider the influence of convective heat transfer from the environment on temperature. Newton’s cooling formula is given as follows:(5)$$q=hA(T-T_{r})$$
in which *Q* represents the heat transfer rate, *h* represents the convective heat transfer coefficient, *A* represents the surface area through which heat is transferred, *T* represents the temperature of the object, and *Tr* represents the temperature of the surrounding environment.

In thermal–mechanical coupling, several common boundary conditions are crucial for accurate simulations. These include displacement boundary conditions, which define how components may move or deform at certain boundaries; force boundary conditions, specifying the forces applied to the system; temperature boundary conditions, setting the temperatures at various system boundaries; heat flux boundary conditions, determining the heat flow into or out of the system at its boundaries; and constraint boundary conditions, which impose restrictions on the movement or deformation of certain parts of the system.

Furthermore, additional conditions must be considered for enhanced accuracy and rationality of the simulation results. These include interface heat conduction conditions, which account for heat transfer across material interfaces; thermal radiation boundary conditions, which model the heat transfer due to radiation; and interface contact conditions, which simulate the thermal interaction at interfaces where components are in contact.

The research focus of thermomechanical coupling typically revolves around the warpage and cracking of packaging structures, primarily induced by coefficient of thermal expansion (CTE) mismatches. The finite element method (FEM) is commonly employed for thermomechanical coupling analysis. When conducting thermal–mechanical FEM simulation, a model is generally established first, and then displacement and temperature boundary conditions are applied for the calculation to obtain the stress distribution. Xue researched the thermal–mechanical simulation of TSV with sidewall scallops, using the element birth and death (B&D) technique [46], as shown in Figure 9a. Compared with the traditional FEM, the B&D technique can simulate the process temperature of each layer of the side wall one by one, providing more accurate results. In the research of Cu protrusions generated by TSV annealing, Jeong considered the nonlinear geometric shape and material properties and the plastic behavior during temperature rise. ABAQUS was used for FEM simulation to obtain the effects of layer number, height, and shape on Cu protrusions [47]. One of the results is shown in Figure 9b.

Packaging warpage is usually simulated using thermomechanical FEM; the process is similar to above. After establishing the model, displacement boundary conditions are set, along with temperature boundary conditions based on the curing temperature, to simulate the warpage deformation. In the latest warpage simulation research, Chiu characterized the coupled chemical–thermomechanical deformation mechanism of a commercial EMC and incorporated it in a finite element model for considering the warpage evolution during the reconstitution thermal processes [48]. Shih studied the warpage of a four-layer stacked-chip wafer. Lee proposed several equivalent material methods to address the difficulties brought by complicated RDLs in warping simulation; the most effective method is shown in Figure 10, which can obtain accurate results [49].

#### 3.1.2. Electro-Thermal Coupling

When an electric current flows through a conductor, heat is inevitably generated. This heat, in turn, influences the conductor’s temperature, which affects the transmission and distribution of the current. There is a reciprocal interaction between the electric field and the thermal field. In advanced packaging, the objective of electro-thermal coupling analysis is to ascertain the steady-state temperature field.

In electro-thermal coupling analysis, the first step is to calculate the Joule heating. Joule heating can be obtained using the current continuity equation and the formula for Joule heating power density. The expressions are as follows:(6)$$\nabla J+\frac{\partial \rho }{\partial t}=0$$
(7)$$p=\frac{{\left|J\right|}^{2}}{\sigma }$$
in which *J* represents the current density, *ρ* represents the charge density, *p* represents the Joule heating power density, and *σ* represents the electrical conductivity.

By considering Joule heating as the heat source, the temperature field distribution can be obtained through the heat conduction equation (Equation (4)).

The effect of the temperature field on the electric field mainly arises from the variation in electrical conductivity with temperature. The calculation expression for electrical conductivity at different temperatures is as follows:(8)$$\sigma =\frac{\sigma_{0}}{1+\alpha \left(T-T_{0}\right)}$$
in which *σ* represents the electrical conductivity, *T* represents the temperature, *T*_0_ represents the reference temperature, *σ*_0_ represents the fixed reference electrical conductivity of the material at the reference temperature (usually at room temperature), and *α* represents the temperature coefficient.

Common boundary conditions in electro-thermal coupling analysis include current boundary conditions, potential boundary conditions, temperature boundary conditions, and convective boundary conditions.

The main factor affecting the reliability of the micro-bump is electromigration. Electromigration is commonly studied through the method of electrical–thermal coupling, aiming to obtain the temperature distribution and current density distribution within the structure. Bump electromigration analysis is typically conducted using the FEM. Firstly, the bump structure is modeled, and current density boundary conditions and temperature boundary conditions are set based on the current equation and heat transfer equation. Tian established an equivalent model of the vertical stacking structure of fan-out wafer-level packages [50]; the simulation loaded the model with an electrical–thermal environment using COMSOL. The model and results are shown in Figure 11. The method also suits Cu pillar interconnects. Li researched the electromigration of Cu pillar bumps under AC conditions. The thermal–electric–mechanical coupling simulation was realized by using the direct coupling element of ANSYS [51].

#### 3.1.3. Electro-Magnetic Coupling

The calculation of electromagnetic coupling can be divided into numerical methods and equivalent circuit methods.

Numerical wave methods utilize techniques such as the Finite Difference Method (FDM) or FEM to solve Maxwell’s Equations (9)–(12) based on geometric shape boundary conditions and material properties. These methods simulate the electric and magnetic fields at various locations. This approach is suitable for cases where the interconnects are non-uniform and long, electromagnetic coupling effects are dominant, or electromagnetic compatibility simulations are required [52].

Maxwell’s equations in their differential form are as follows:(9)$$\frac{\partial B}{\partial t}+\sigma_{m}H+\nabla \times E=-M_{s}$$
(10)$$\frac{\partial D}{\partial t}+\sigma_{e}E-\nabla \times H=-J_{s}$$
(11)∇⋅D=ρ
(12)∇⋅B=0
in which *Js* represents applied electricity, *Ms* represents magnetic current densities, *σ_e_* represents electrical conductivity, and *σ_m_* represents magnetic conductivity.

As mentioned in the previous section, electrical conductivity is temperature-dependent. Therefore, in interconnect structures, electromagnetic fields and temperature fields interact with each other, allowing electromagnetic–thermal coupling to be achieved using Maxwell’s equations, the heat conduction equation, and boundary conditions.

The equivalent circuit method involves converting various conductors and electrolytes in the structure into circuit elements such as resistors, capacitors, inductors, and transmission lines. This method streamlines the modeling process and facilitates rapid calculations through conventional circuit analysis techniques. It is beneficial for analyzing effects such as near-field crosstalk, transmission line propagation, reflection, and switch noise.

Among these elements, resistance is susceptible to temperature. Thus, the variation in resistance with temperature can be considered as a coupling term in electromagnetic–thermal coupling.

By extracting boundary conditions, synergistic simulation between the field and circuit can be achieved, thereby improving the accuracy of the results. In recent research, Zhang proposed a method to establish a coupling relationship between the electromagnetic equations and circuit equations using port voltages, port currents, and port elements to construct the electric field [53].

In terms of electrical performance, TSV serves as the carrier of electrical connections and the main research object for electrical reliability. The electrical performance of TSV structures is influenced by factors such as temperature and frequency. Therefore, electro-thermal coupling analysis is necessary.

The equivalent circuit method is commonly used to research TSV’s electrothermal effect. Firstly, RLCG parameters are extracted based on the design parameters of TSV. Then, thermal simulation is performed on the structure, usually using the finite element method to obtain temperature distribution. Due to the temperature dependence of material parameters such as conductivity, the obtained temperature distribution is used to update the electrical parameters of TSV, and finally calculate losses and crosstalk to achieve thermoelectric coupling. Min combined the equivalent circuit and equivalent thermal network models to propose a coaxial TSV electro-thermal model, as shown in Figure 12. The thermal dissipation was obtained using the equivalent circuit method, and the thermal response was obtained using the equivalent thermal network. The fixed temperature boundary condition is represented as a constant voltage source, and the power dissipation is defined as a current source [54]. Regarding the insertion loss caused by mutual inductance coupling between neighboring TSVs in a TSV array, Wang proposed an electro-thermal model based on partial element equivalent circuit method to predict the resistance and inductance [55].

Numerical methods are more suitable for the electric heating problem of the power supply network (PDN), TSV arrays, or 3D ICs. The numerical method takes the Maxwell equations and heat conduction equations as the governing equations, and the two are coupled through the temperature dependence of power losses and material parameters. Li developed an FEM-based solver to investigate the electrothermal characteristics of a two-chip power delivery network (PDN) structure, circularly solving the wave equation for a full-wave electromagnetic analysis and the heat conduction function for the thermal analysis [56]. He developed a 3D integrated chip flow heat transfer model under the action of electrothermal coupling and proposed a full-wave electromagnetic simulation model of the signal interconnect components with the same structural parameters; a coupled analysis of electrothermal effects and fluid heat transfer models was achieved by COMSOL Multiphysics [57].

Based on the above content, Table 1 is listed below to summarize the coupling field classification, research objects, and simulation method.

### 3.2. Current Status of Reliability Research on Multi-Field Coupling in Advanced Packaging

The reliability issues of advanced packaging are extensively discussed in Section 2, which can be summarized as structural failures caused by thermal issues, electromigration, and signal transmission crosstalk due to electrothermal coupling. Therefore, research on advanced packaging multi-field coupling reliability mainly focuses on these aspects.

Most of the research on advanced packaging reliability focuses on the reliability of interconnect structures (bumps, TSVs, RDLs). As the connection point of each layer in the packaging, bumps are also the weakest structure in the packaging and are most prone to failure, determining the device’s reliability. By analyzing the thermo-mechanical coupling, the distribution of temperature and stress fields within the package can be obtained. This analysis helps to predict the fatigue life of the bumps and optimize material and structural designs. Sham considered the interaction between temperature and thermal stress in FC and found that the temperature distribution within the package may vary due to the presence of interfacial defects, as shown in Figure 13a. With increasing interfacial length, the stress adjacent to the interfacial region in the solder gradually changes [58], as shown in Figure 13b. Shantaram conducted transient thermal simulations to extract temperature data, which were then used as loading conditions in subsequent thermo-mechanical analyses. Chip areas are selected as the heat zone, as shown in Figure 14a. The fatigue process of FC solder bumps was simulated, and the influence of the bottom encapsulant’s glass transition temperature (*Tg*) on bump fatigue was studied [59], as shown in Figure 14b.

The issue of electromigration in bumps is a significant focus in research on the reliability of advanced packaging. Through multi-field coupling analysis, the distribution of current density and temperature in the bumps can be calculated to predict the electromigration lifetime. This information can be used to select more suitable materials, optimize the dimensions, shapes, and layouts of the bumps, and reduce the risk of electromigration. Lai was the first to apply electrothermal coupling analysis to predict the electromigration lifetime of FC structures. By applying a constant current to a series of daisy-chain solder bumps, the calculated current crowding and Joule heating effects were integrated into Black’s equation [60]. This method was then used to study the Joule heating and current crowding effects in three types of FC bumps [61], as shown in Figure 15a. Lai also conducted an electrothermal coupling analysis of current density and temperature distribution in FC composite solder bumps made of Sn-Ag-Cu alloy [62], as shown in Figure 15b. Chen proposed a new semi-analytical transient stress analysis method to study the influence of thermal migration (TM) on electromigration (EM) induced by Joule heating [63]. Fan conducted electro-thermal–mechanical coupling analysis on cylindrical, hourglass, and barrel-shaped bumps, obtaining the current density distribution, temperature distribution, and equivalent stress distribution for the three bump shapes. It was found that cylindrical bumps were a good choice for improving the resistance to electromigration in copper pillar interconnect structures [27].

WLP is also an advanced packaging technology. The reliability research of WLP mainly focuses on the reliability of solder joints and the warpage during the processing, and thermal–mechanical finite element analysis is a commonly used method to study these issues. In the past, the impact of structure on WLP reliability has received attention [64,65]. Fan conducted finite element modeling on four types of WLP structures, as shown in Figure 16a; the results show that balls on polymer WLP and copper post WLP have good thermal mechanical reliability performance [20]. Recently, it was shown that a novel fan-out WLP with a TSV array interposer layer has higher electrical performance and easier customization. However, warpage formed from the molding process is still an essential issue. Yu presents a process-dependent simulation methodology based on nonlinear FE analysis and the element death–birth technique to explore the warpage of the novel FOWLP. The optimal design using RSM analysis can decrease the maximum smiling/concave warpage value and frowning/convex warpage value by 33% and 36%, as shown in Figure 16b [66].

In terms of the FOWLP, mentioned above, another research focus is the reliability of RDLs. The failure of the RDL copper line is the most concerning. In the latest research, Han researched the von Mises stress of RDLs of different designs, finding that the gap width was the most critical factor affecting the stress of a silicon-based fan-out package [67], as shown in Figure 17a. Liang discovered that the copper line of an RDL with a 45° bending structure exhibited better resistance to electromigration compared with the straight-line one [24]. Figure 17b shows the stress distribution analysis result, which proved that the straight-line Cu RDL is more vulnerable to mechanical deformation.

RDLs also affect the warpage of packaging structures. Lin presented a novel fan-out WLP with the RDL-first method. The FEM was used to optimize the warpage control [68]. Shih introduced an FEM model applied to investigate the thermomechanical properties of a multilayer RDL, finding that a smaller overall size of the fan-out interposer is beneficial in reducing the warpage [69]. Figure 18a shows the warpage (z) versus PCB thickness (x) and interposer size (y). Lee used the FEM to simulate the warpage of fan-out panel-level packaging (FOPLP) with RDLs and dielectric layers during fabrication, and the result is shown in Figure 18b [70].

As electronic packaging technology developed from 2D to 2.5D and 3D, the implementation of TSVs and micro-bumps has enabled vertical interconnects of integrated circuits. However, TSVs face challenges in electrical performance and mechanical properties under multi-physical fields, leading to increased attention to reliability research of TSV structures. Conducting multi-field coupling analysis on TSV structures and 3D integrated systems helps to identify potential failure and optimize the structures to enhance reliability.

In terms of TSV mechanical reliability, Cheng proposed a 3D IC thermal–mechanical coupling field simulation method based on an equivalent homogenized modeling approach and sub-modeling technique. This method significantly improves the accuracy of local temperature and warpage deviation compared to traditional methods [71], as shown in Figure 19a. Ni conducted finite element simulations to investigate the mechanical performance and lifespan of TSVs under electro-thermal coupling. Through an orthogonal experimental analysis, they validated the high reliability of TSVs filled with carbon nanotubes (CNTs) [72], as shown in Figure 19b. Wang proposed an improved hybrid time-domain finite element method and used it to simulate the transient electro-thermal–mechanical response of multilayer TSVs under periodic voltage pulses. The results, as shown in Figure 19c, revealed the sensitivity of the outcomes to changes in the surrounding silicon oxide isolation thickness [73]. He studied the electro-thermal coupling effects of TSVs embedded in micro-pin-fin structures of 3D ICs. The analysis examined the influence of parameters such as micro-pin-fin and TSV layout, TSV diameter, oxide thickness, and pin-fin structure on the electro-thermal coupling behavior [61]. Figure 19d illustrates the impact of TSV diameter.

In terms of TSV electrical performance and system signal integrity, Vincenzo proposed a dynamic electro-thermal macro modeling technique for signal integrity analysis. This method extracted the dynamic electro-thermal equivalent of multi-chip stack structures and estimated the thermally induced propagation delay through circuit simulations using SPICE [75]. Wang studied the electro-thermal effects in TSVs and investigated the influence of temperature on the parasitic resistance and capacitance of TSVs by combining copper characteristics with an equivalent circuit model [55], as shown in Figure 20a. Yang conducted an electro-thermal–mechanical coupling analysis of the intermediate layer in TSVs, analyzing the electrical, thermal, and mechanical performance of the intermediate layer. Figure 20b shows the comparison of S parameters for TSV interposers before and after deformation, revealing that the structural deformation caused by electro-thermal coupling reduces the efficiency of electrical signal transmission [76].

SiP integrates multiple functional components into the same package to form an electrical system. However, the 2D, 2.5D, or 3D packaging structure of SiP introduces the complexity of thermal management and electrical transmission, so it is necessary to carry out multi-field coupling analysis on the packaging structure. Touati used the Foster model to predict the temperature of inter-layers that constitute the SiP, as is shown in Figure 21a, further predicting the stress and thermo-mechanical behavior of systems comprising 3D-ICs [77]. Chen studied the thermo-mechanical reliability of the RF SiP module based on an LTCC substrate, as shown in Figure 21b, focusing on the reliability of the heat reflow process, the operating state, and fatigue of second-level solder joints [78]. Tang used the FEM model to simulate the thermal stress distribution in the stacked die of a four-tier die-stacked SiP during thermal cycling, and the Taguchi method was used for optimal design [79]. Hsieh conducted a simulation on RF impedance matching, power integrity, and thermal distribution with a complete Wi-Fi SiP module [80]. Figure 22 shows the result.

## 4. Conclusions and Future Perspectives

This paper provides an overview of the development of advanced packaging, the reliability issues under various physical fields, and the multi-field coupling methods and research progress in advanced packaging. The summary and future perspectives of the research on multi-field coupling reliability in advanced packaging are outlined as follows:(1)Interconnect technology plays a critical role in advanced packaging as it serves as both an electrical transmission channel and the most susceptible area for failures within the packaging. Presently, research on multi-physics coupling reliability in advanced packaging primarily focuses on the reliability of interconnects, including bumps, RDLs, and TSVs. This research particularly emphasizes the coupling effects among electrical, thermal, and mechanical fields.(2)With the continuous advancement of electronic packaging technologies, such as 2.5D and 3D packaging, WLP, etc., these new generation packaging technologies will face more complex physical fields and reliability challenges. Future research will concentrate on multi-field coupling analysis and the optimization of reliability for these emerging technologies.(3)The future trend in multi-physics coupling research is to integrate models of different physical fields more tightly to describe the effects of multi-field coupling accurately. However, solving multi-physics problems is highly complicated due to multi-scale phenomena and nonlinearity. Currently, commonly used numerical methods include FEM, FDM, and other numerical computation methods. The computational cost is high for complex bidirectional and transient coupling, resulting in lower efficiency. Many researchers have explored improved optimization algorithms, and in the future, machine learning and artificial intelligence algorithms are expected to play a role in multi-field coupling computations.

## Figures and Tables

**Figure 1 micromachines-15-00422-f001:**
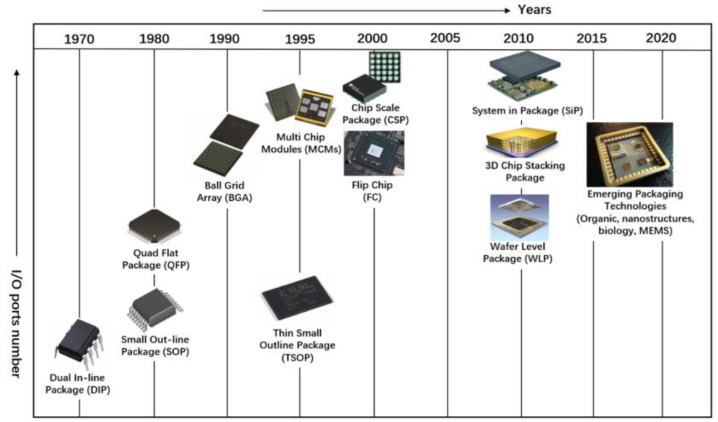
Development schedule of electronic packaging forms [1].

**Figure 2 micromachines-15-00422-f002:**
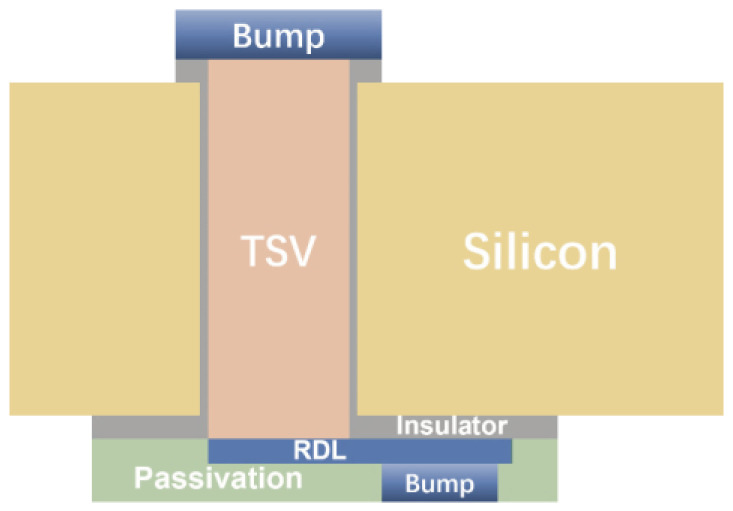
Interconnect forms in advanced packaging.

**Figure 3 micromachines-15-00422-f003:**
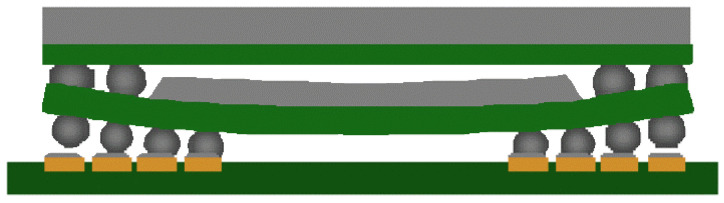
Open joints due to warpage of Package on Package (PoP) [11].

**Figure 4 micromachines-15-00422-f004:**
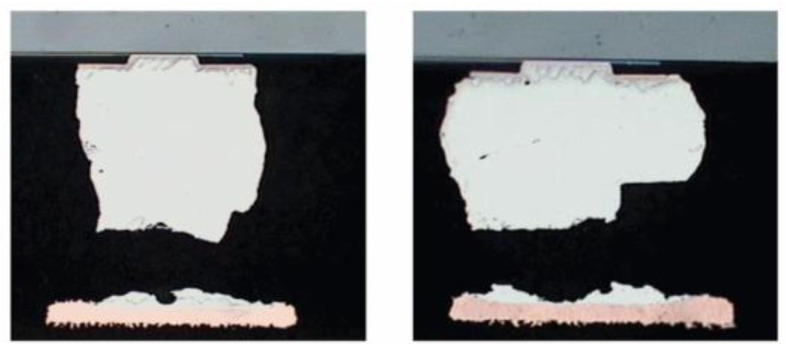
Crack images of FC bumps without underfill under thermal cycling [18].

**Figure 5 micromachines-15-00422-f005:**
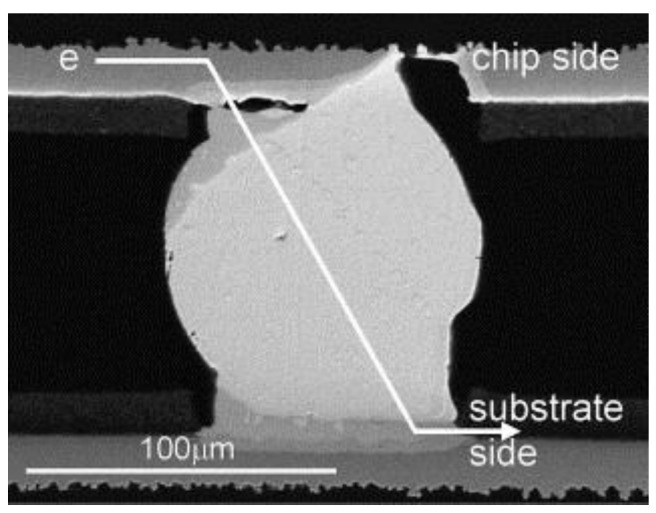
Bump cross-section under electromigration, with arrows representing the direction of electron flow [30].

**Figure 6 micromachines-15-00422-f006:**
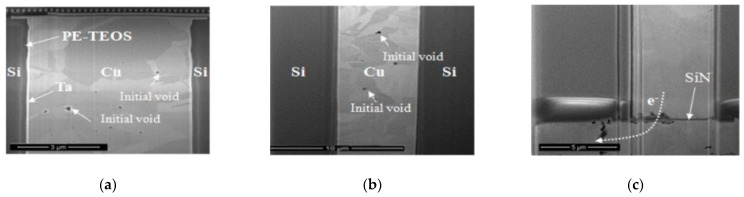
(**a**) TSV top electromigration hole. (**b**) TSV middle electromigration cavity. (**c**) TSV bottom electromigration hole [35].

**Figure 7 micromachines-15-00422-f007:**
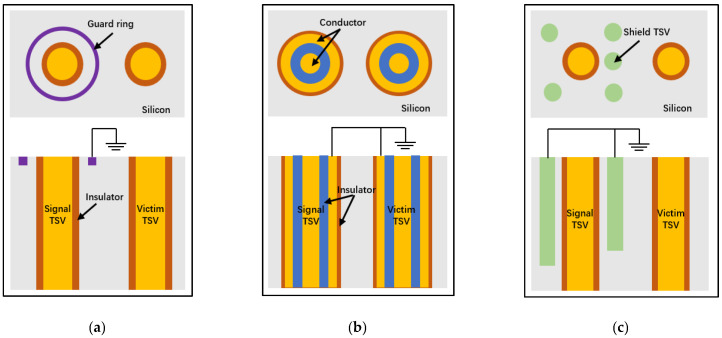
The top and cross-section views of different TSV structures. (**a**) TSVs with a guard ring [43]. (**b**) Coaxial TSV structure [40]. (**c**) A TSV structure with a ground interconnect array [44].

**Figure 8 micromachines-15-00422-f008:**
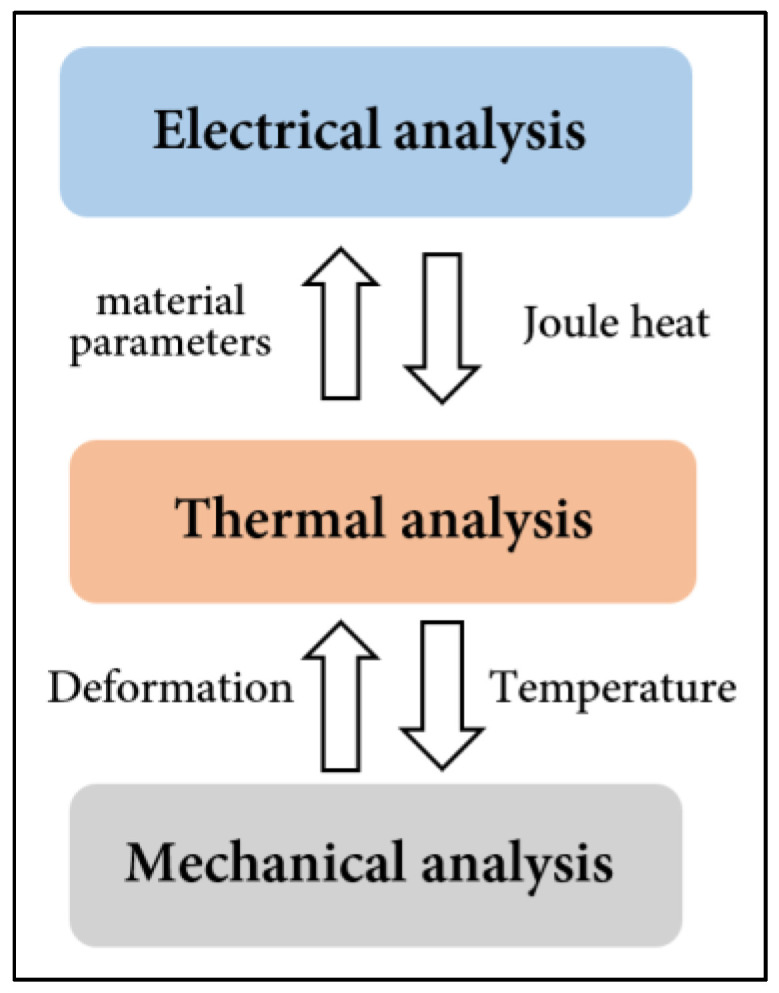
Multi-physical-field coupling relationship.

**Figure 9 micromachines-15-00422-f009:**
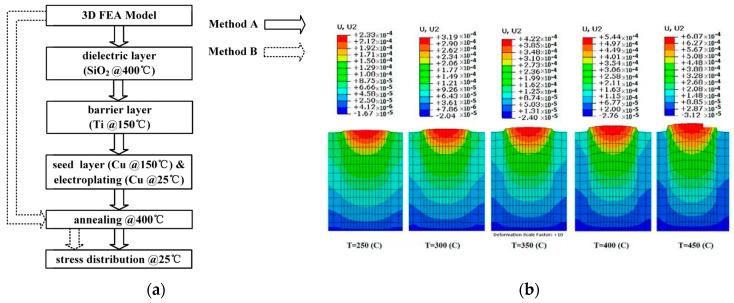
(**a**) Simulation flows for FEA with B&D (Method A) and without B&D (Method B) [46]. (**b**) Vertical deformation of the cylindrical TSV shape [47].

**Figure 10 micromachines-15-00422-f010:**
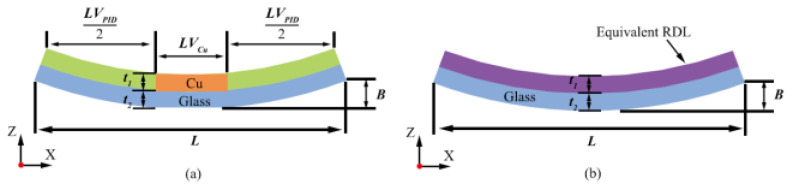
Description diagram of the modified Timoshenko bi-material approach [49]. (**a**) Original RDL pattern. (**b**) Equivalent RDL material.

**Figure 11 micromachines-15-00422-f011:**
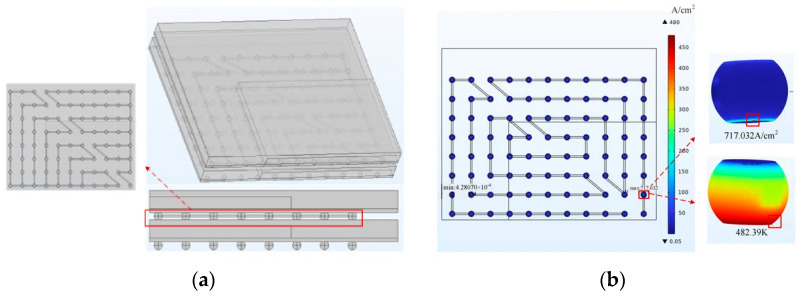
(**a**) Three-dimensional diagram of 1/4 micro-component model. (**b**) Cloud plot of the results at the current aggregation [50].

**Figure 12 micromachines-15-00422-f012:**
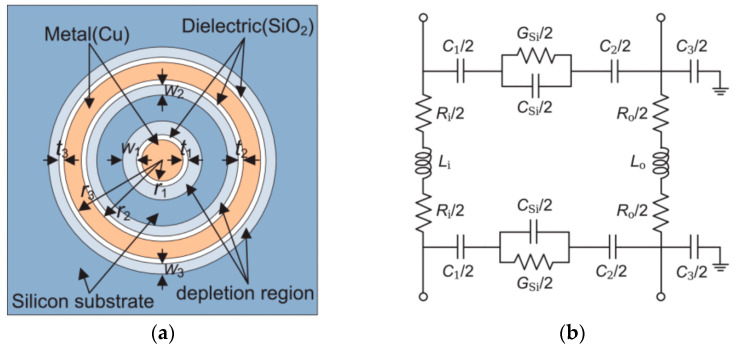
(**a**) Cross-sectional view with the depletion regions shown. (**b**) Electrical equivalent circuit model [54].

**Figure 13 micromachines-15-00422-f013:**
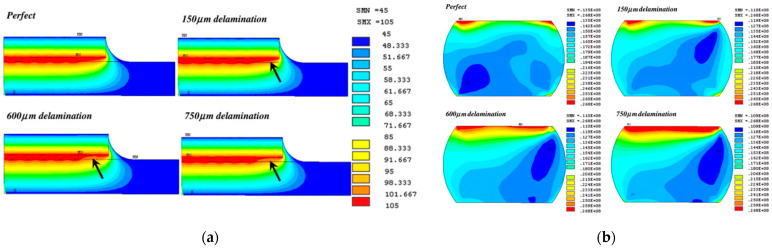
(**a**) Temperature distributions in the package with different delamination lengths. (**b**) Von Mises stress distributions in the solder bump with varying delamination lengths [58].

**Figure 14 micromachines-15-00422-f014:**
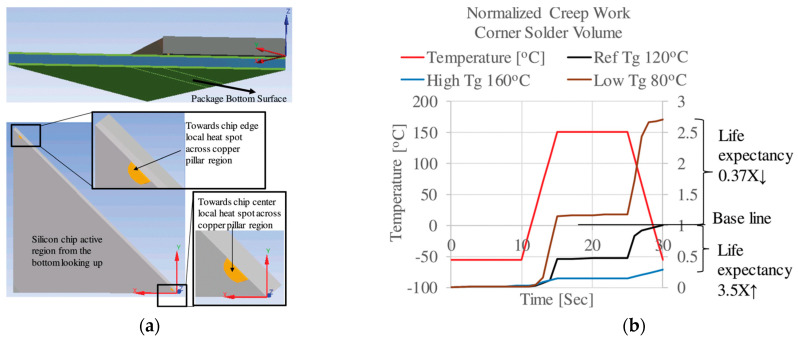
(**a**) Chip areas as heat zones (3 cases). (**b**) Effect of underfill *Tg* on bump fatigue life [59].

**Figure 15 micromachines-15-00422-f015:**
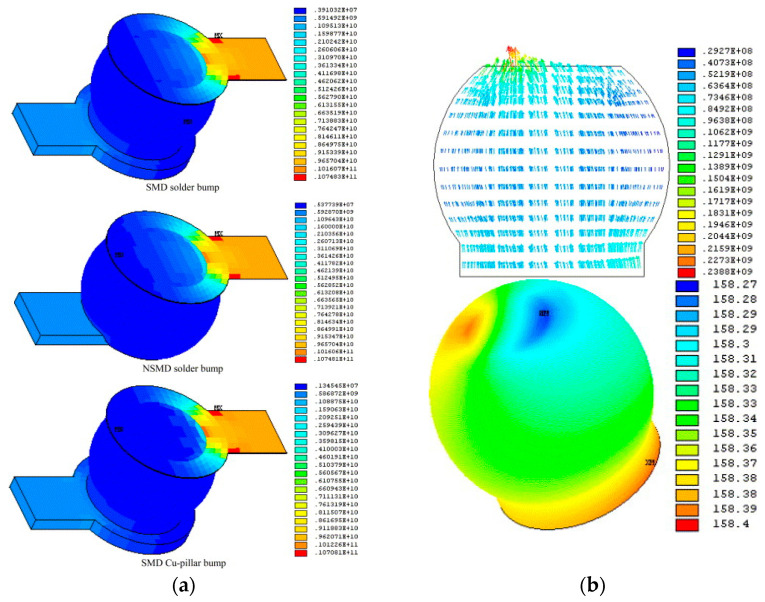
(**a**) Current density contours on V2+ trace-and-bump systems under 0.64 A/127 °C (unit: A/m^2^) [61]. (**b**) Current density distribution and temperature distribution in Sn-Ag-Cu bump E under 0.5 A/150 °C [62].

**Figure 16 micromachines-15-00422-f016:**
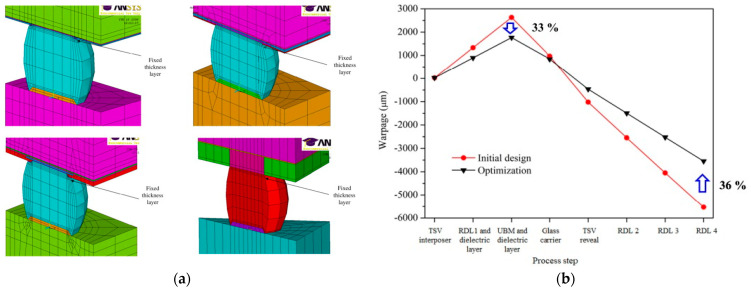
(**a**) Finite element models of 4 WLP structures [20]. (**b**) Maximum negative warpage value and maximum positive warpage value can be significantly reduced by 33% and 36% through optimal design using RSM analysis [66].

**Figure 17 micromachines-15-00422-f017:**
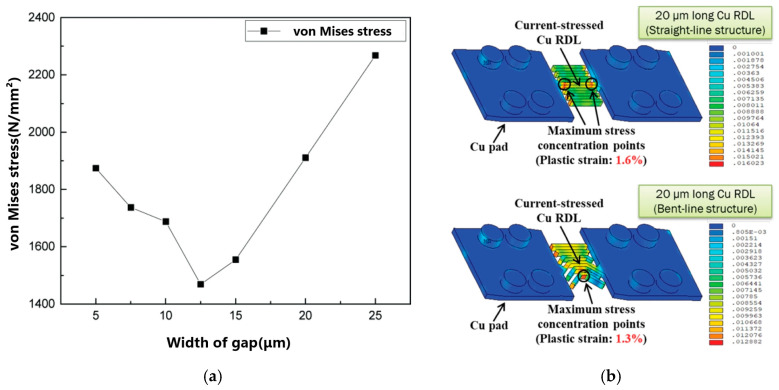
(**a**) The impact of the height of the chip and substrate on von Mises stress [67]. (**b**) Stress distributions in the 20 μm long Cu RDLs with straight-line and bent-line (45°) structures [24].

**Figure 18 micromachines-15-00422-f018:**
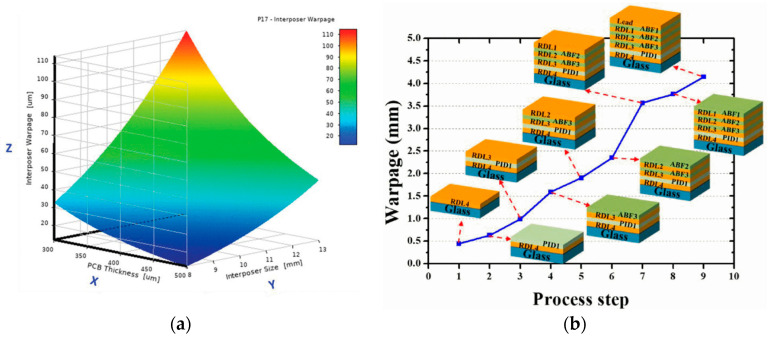
(**a**) Fan-out interposer warpage (z) versus PCB thickness (x) and interposer size (y) [69]. (**b**) Warpage during the manufacturing process in simulation [70].

**Figure 19 micromachines-15-00422-f019:**
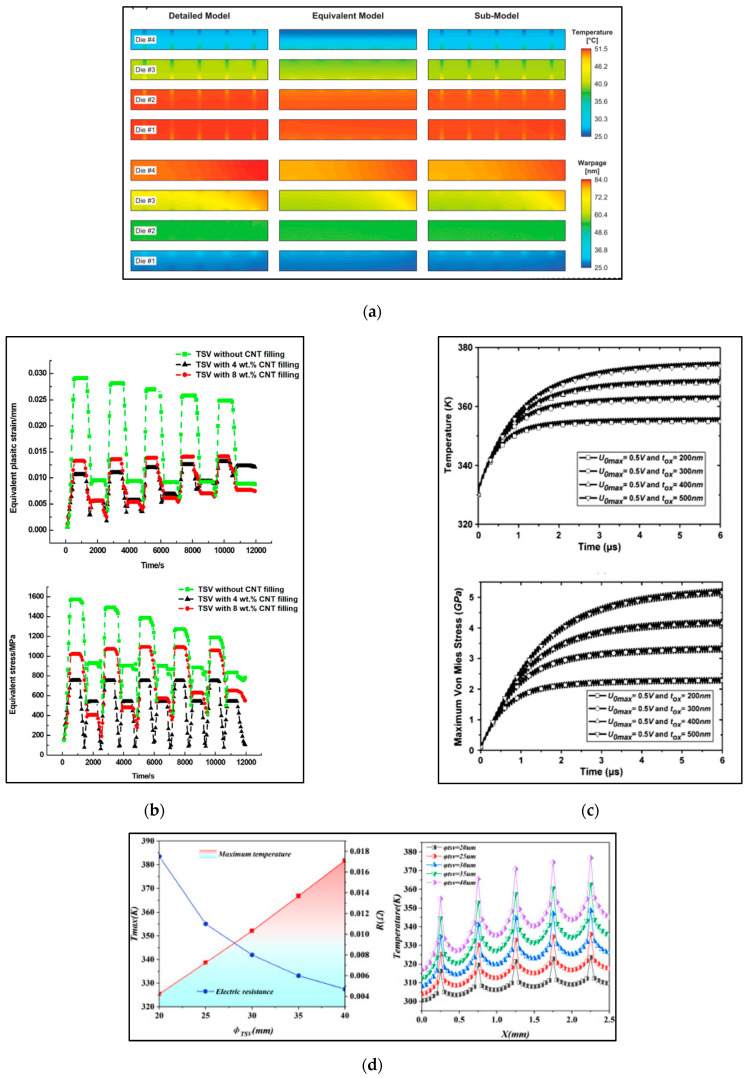
(**a**) The temperature and warpage distributions of the section selected along the diagonal of the 3D model [71]. (**b**) Equivalent plastic strain and equivalent stress for TSVs with different CNT filling degrees [72]. (**c**) Transient temperature and thermal stress responses of the three-layered W/poly-Si TSV with a trapezoidal voltage pulse applied [73]. (**d**) Effect of different TSV diameters on temperature and electrical resistance, and temperature distribution at the midline position [74].

**Figure 20 micromachines-15-00422-f020:**
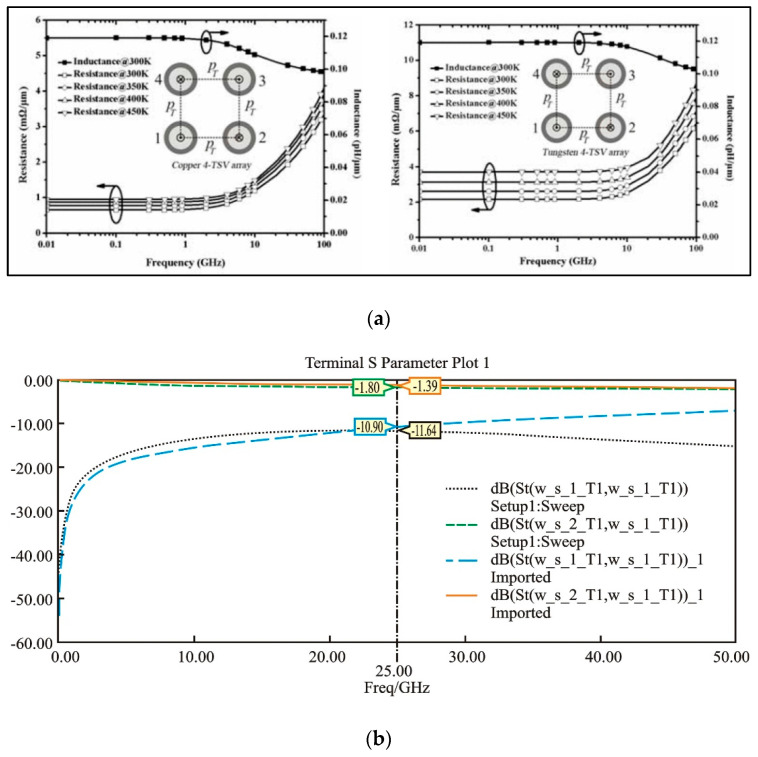
(**a**) Per-unit-length resistances and inductances of a copper 4-TSV array and a tungsten 4-TSV array at different ambient temperatures [55]. (**b**) Comparison of S parameters of TSV interposer before and after deformation [76].

**Figure 21 micromachines-15-00422-f021:**
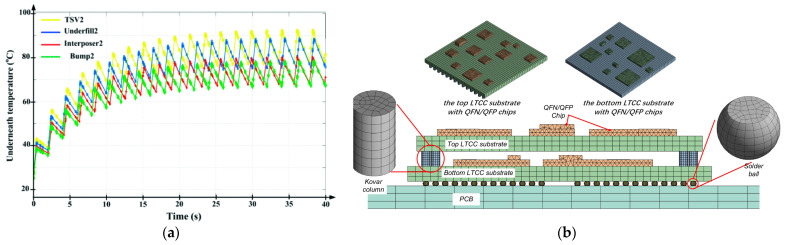
(**a**) Temperature distribution of the layers obtained by the Foster model [77]. (**b**) Schematic of the 3D finite element model [78].

**Figure 22 micromachines-15-00422-f022:**
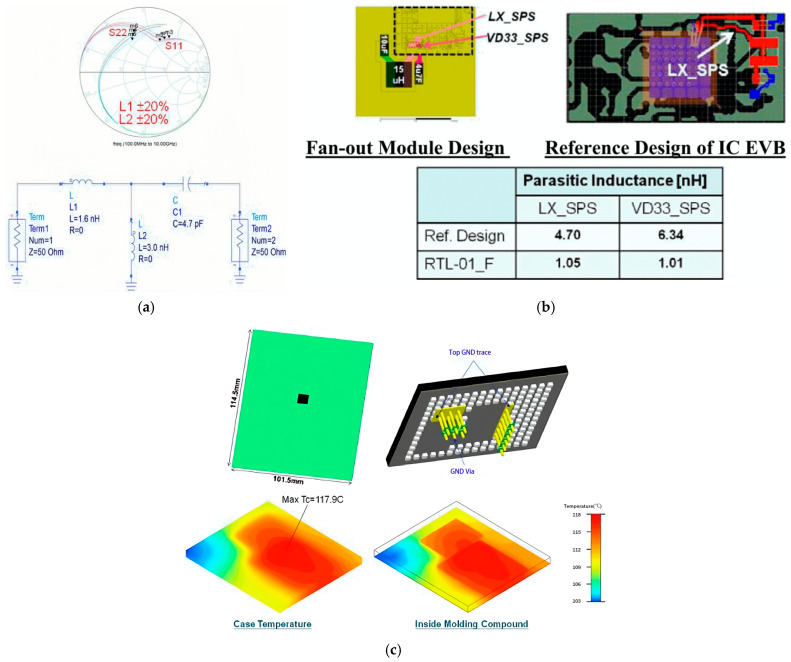
(**a**) RF impedance matching analysis. (**b**) Parasitic inductance analysis. (**c**) Thermal analysis [80].

**Table 1 micromachines-15-00422-t001:** Advanced packaging multi-field coupling simulation methods.

Research Object	Coupled Field	Simulation Method
Electromigration of interconnects	Electro-thermal	FEM (based on the current equation and heat conduction equation)
Electrical performance of packaging structure or interconnects	Electro-thermal Electro-magnetic	FEM (based on the current equation and heat conduction equation); Equivalent circuit method; Full wave solver (based on Maxwell’s equations)
Packaging thermal stress, warpage, crack, etc.	Electro-thermal Thermo-mechanical	FEM (based on the current equation, Heat conduction equation, and strain equation)

## Data Availability

The data presented in this study are available on request from the corresponding author.

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
