# Peer review of "Research on the Reliability of Advanced Packaging under Multi-Field Coupling: A Review"

_micromachines, 2024, doi:10.3390/mi15040422_

Round 1
Reviewer 1 Report
Comments and Suggestions for Authors
It is a pleasure to respond to this review, in which the reliability of advanced packages under multi-physical field coupling is addressed. Overall the review is not bad, but there are still some issues that need to be addressed. First of all, in this review, I think the content of chapter 3 is the most critical part, but it starts with a lot of principle content that should belong to the second part. Adding some experimental pictures like in the second half would make this paragraph more integrated with Chapter III. In addition, advanced packaging includes not only Flip Chip and 25D / 3D packaging, but also technologies such as SIP, WLP, and RDL, which could be added to the content. Finally, the language of the review needs to be further optimized, so please try to revise the grammar, word format and proofread carefully before submission. For example, there may be some problems with Table 1.
Author Response
Thank you for your valuable comments. In response to your review comments, please see the attachment for our response. Thank you! The specific modified parts in the manuscript have been highlighted in red. Please review them again and provide valuable comments. Thank you very much!

Reviewer 2 Report
Comments and Suggestions for Authors
This paper focused on the current status of multi-field coupling reliability of interconnects in advanced packaging. However, the latest publications are not well organized in this review. In the section of "Advanced Packaging Multifield Coupling Research Progress", some equations regarding thermal-mechanical coupling calculation, heat transfer, electrical conductivity, and the Maxwell's equations were described in detail, but the specific examples are far more less to give readers a clear picture of research status in this field. In addition, these simulations did not reflect the characteristics in advanced packaging. The graphs in this manuscript are not so representative.
Comments on the Quality of English LanguageMany English grammar mistakes still existed, and the English writing needs to be polished. For example, in abstract, "With the development of Moore's Law reaching its limits, advanced packaging technologies represented by FC, WLP, SiP, and 3D packaging have received significant attention and research. " “research” is redundant. "While advanced packaging has made breakthrough progress in achieving high performance, miniaturization, and low cost, the smaller thermal space and higher power density have created complex physical fields such as electricity, heat, and force." "progress" is unnecessary. Although these sentences can be understood, the expression is not aligned with the English habits. So, the English writing needs to be revised by native speaker or the language agency.
Author Response
Thank you for your valuable comments. In response to your review comments, please see the attachment for our response. Thank you! The specific modified parts in the manuscript have been highlighted in red. Please review them again and provide valuable comments. Thank you!

Reviewer 3 Report
Comments and Suggestions for Authors
This review paper is about the reliability of advanced packaging based on multifield coupling simulation.
It should be interesting to get some ideas on reliability of advanced packaging with coupled field simulation methods.
However, it is not clear to find the usefulness of the coupled filed situation for advanced electronic packaging in the submitted manuscript.
Here are some comments and questions.
1. Thermal cycling reliability of C4 solder should consider the creep of the solder.
2. Coupled field simulation part deals with well-known theory and already used in the existing system like mems.
3. Please present the coupled field simulations with the approach detailed with concept, meaning, uniqueness, etc.
4. Summarizing the different approach in the coupled filed approaches in a table should be useful.
5. The current examples in the submitted manuscript is not new ones but seems already well-known. Please focus to the novelty of the coupled filed reliability simulation as the paper title.
Author Response

(The authors gave the same response as above.)

Round 2
Reviewer 1 Report
Comments and Suggestions for Authors
The reviewer has already read the reply and the revisions you made to the review, which are very much improved and enriched, but here are some more revisions to be made.
First of all, in the first comment, I mean that there is a lack of citation and pictures in 3.1.1 Thermal coupling, 3.1.2 Electro-thermal coupling, 3.1.3 Electromagnetic coupling, and this time, you have revised the citation of the substantial progress content all added in the back of 3.1.3 Electromagnetic coupling, which looks very messy. You should put the corresponding multi-physics field coupling content and pictures in the corresponding subsection locations to make these three subsections look less off-principle and thus distinguish them from Chapter II.
Second, more detailed explanations and pictures are needed for the newly added RDL study content, which can be done by following the modifications to the wafer-level package.
Again, it would enrich the review to have a study of the package structure as a whole, such as the system level packaging mentioned in the first comment.
Author Response
First of all, in the first comment, I mean that there is a lack of citation and pictures in 3.1.1 Thermal coupling, 3.1.2 Electro-thermal coupling, 3.1.3 Electromagnetic coupling, and this time, you have revised the citation of the substantial progress content all added in the back of 3.1.3 Electromagnetic coupling, which looks very messy. You should put the corresponding multi-physics field coupling content and pictures in the corresponding subsection locations to make these three subsections look less off-principle and thus distinguish them from Chapter II.
----- The reviewer is very professional. The overall structure of subsection 3.1 has been modified, and the multi field coupling method for different structures has been placed behind the corresponding section.
Second, more detailed explanations and pictures are needed for the newly added RDL study content, which can be done by following the modifications to the wafer-level package.
----- The reviewer is very professional. From line 573 to 592, we added more content and pictures to this section, so as to better integrate RDL with the WLP mentioned earlier.
Again, it would enrich the review to have a study of the package structure as a whole, such as the system level packaging mentioned in the first comment.
----- The reviewer is very professional. System level packaging includes 2D, 2.5D and 3D Packaging, so we didn't make a good distinction before. In the end of the section 3.2, some research on multi field coupling of SiP structure is added.

Reviewer 2 Report
Comments and Suggestions for Authors
1. Line457, "After establishing the model, set displacement boundary conditions and set temperature boundary conditions based on the curing temperature to simulate the warpage deformation". This sentence does not conform to grammar.
2. Many "figure" in the text should be "Figure" or Fig.
3. There are glitches on the edges of the graphic in Figure 3.
4. Fig. 12(a) was flipped.
Comments on the Quality of English LanguageThe grammar mistakes or typos still existed, and need mor polishing.
Author Response
1.Line457,"After establishing the model, set displacement boundary conditions and set temperature boundary conditions based on the curing temperature to simulate the warpage deformation". This sentence does not conform to grammar.
----- The reviewer is very professional. The original content is modified as “After establishing the model, displacement boundary conditions are set, along with temperature boundary conditions based on the curing temperature, to simulate the warpage deformation.”
- Many "figure" in the text should be "Figure" or Fig
----- The reviewer is very professional. “figure” in the text has been modified as “Figure”.
- There are glitches on the edges of the graphic in Figure 3
----- This reviewer is very detailed and professional. The quality of Figure 3 has been improved.
- Fig. 12(a) was flipped.
----- This reviewer is very detailed and professional. The Figure has been modified.
5.The grammar mistakes or typos still existed, and need more polishing
----- The reviewer is very professional. The author has checked and revised the English writing by sentence, and the grammar mistakes have also been checked by software.

Reviewer 3 Report
Comments and Suggestions for Authors
Thank you for the revised paper.
Author Response
Thank you for your review comments. We have made further revisions.
Round 3
Reviewer 1 Report
Comments and Suggestions for Authors
The author has revised the manuscript. After careful review, the reviewers found the article generally acceptable for publication.